# Synthesis of Ti_3_SiC_2_ Phases and Consolidation of MAX/SiC Composites—Microstructure and Mechanical Properties

**DOI:** 10.3390/ma16030889

**Published:** 2023-01-17

**Authors:** Jaroslaw Wozniak, Mateusz Petrus, Tomasz Cygan, Boguslawa Adamczyk-Cieślak, Dorota Moszczyńska, Andrzej Roman Olszyna

**Affiliations:** Faculty of Material Science and Engineering, Warsaw University of Technology, 02-507 Warsaw, Poland

**Keywords:** sintering, composites, mechanical properties, SiC, MAX phases

## Abstract

The article describes the Ti_3_SiC_2_ powder synthesis process. The influence of the molar ratio and two forms of carbon on the phase composition of the obtained powders was investigated. The synthesis was carried out using a spark plasma sintering (SPS) furnace. In addition, using the obtained powders, composites reinforced with SiC particles were produced. The obtained results showed no effect of the carbon form and a significant impact of annealing on the purity of the powders after synthesis. The composites were also consolidated using an SPS furnace at two temperatures of 1300 and 1400 °C. The tests showed low density and hardness for sinters from 1300 °C (maximum 3.97 g/cm^3^ and 447 HV5, respectively, for composite reinforced with 10% SiC). These parameters significantly increase for composites sintered at 1400 °C (maximum density 4.43 g/cm^3^ and hardness 1153 HV5, for Ti_3_AlC_2_—10% SiC). In addition, the crack propagation analysis showed mechanisms typical for granular materials and laminates.

## 1. Introduction

It is well known that metallic materials are characterized by plasticity, good electrical and thermal conductivity, and are easy to machine. However, the operating temperature is a significant limitation of the use of metals. At high temperatures, the mechanical properties of metals are rapidly decreasing. On the other hand, ceramic materials are characterized by resistance to high temperatures, high chemical resistance, and high hardness [1,2]. However, their significant limitation is the brittleness and difficulty in shaping these materials. For many applications, the ideal solution would be to create a material that combines the features of these two material groups. Most often, this is achieved by manufacturing composites on a ceramic matrix to improve fracture toughness and on a metallic matrix to improve mechanical properties and stability at elevated temperatures. Another group of materials combining the features of metals and ceramics is the so-called MAX phases. Their name is related to their composition M_n+1_AX_n_, where M—light transition metal, e.g., Ti, Nb, V; A (13 or 14 group metals), and X is carbon or nitrogen. The MAX phases show a layered structure where between M-X layers is located a metallic layer [3,4]. Due to covalent–metallic–ionic bonds, the MAX phases are characterized, like ceramics, by good resistance to oxidation, thermal stability, relatively high melting point, and high strength properties. Like metals, they display good electrical and thermal conductivity, good thermal shock resistance, and abrasion resistance and are relatively easily mechanically machined [5,6]. Such a diverse set of properties of the MAX phases predisposes them to many applications, i.e., protective barriers, materials working at elevated temperatures, abrasion-resistant materials, and many others [7,8,9,10,11].

Despite the many advantages of MAX phases, these materials are characterized by relatively low hardness and wear resistance. In addition, they decompose at higher temperatures. Barsoum et al. [12] showed the decomposition of Ti_3_SiC_2_ at temperatures above 1000 °C and the formation of TiO_2_ on the surface and a mixture of TiO_2_ and SiO_2_ inside as a result of inward diffusion of oxygen and outward diffusion of titanium and carbon. Similarly, Low et al. [13] showed a slow decomposition of Ti_3_SiC_2_ into non-stoichiometric TiC_x_ and Ti_5_Si_3_C_x_ at 1200 °C in a vacuum and a significant acceleration of decomposition above 1500 °C. They proved that the stability of Ti_3_SiC_2_ is dependent on the oxygen partial pressure of the annealing atmosphere within the furnace. To improve the mechanical properties and thermal stability of the MAX phases, composites with the addition of a ceramic phase are produced. In the case of Ti_3_SiC_2_, the obvious choice, due to its thermodynamic compatibility, is to use SiC as the reinforcing phase. SiC was added in the form of particles or fibers. Tong et al. [14] produced Ti_3_SiC_2_ matrix composites containing 20 vol% SiC, which were then hot pressed at 1600 °C. The authors showed the hardness of composites of about 10.8 GPs and an increase in fracture toughness compared to unreinforced sinters. Similar results were obtained by Radhakrishnan et al. [15] for Ti_3_SiC_2_—14 vol% SiC composites. They obtained hardness and fracture toughness of 8.9 GPa and 9.1 MPa*m^0.5^, respectively, and these values were higher than for the reference sample. A different approach to the production of Ti_3_SiC_2_/SiC composites was presented by Zhang et al. [16]. The composites were produced by reactive sintering by spark plasma sintering (SPS) method from pure powders. As a result of the sintering, composites reinforced with 20 vol.% SiC were obtained. Compared to the literature data for pure Ti_3_SiC_2_ sinters, they received an increase in hardness and a decrease in bending strength. Additionally, in the case of composites reinforced with SiC fibers, it was possible to produce composites characterized by high density, good connection at the matrix–fiber interface (without degradation of the fiber at the interface), and better creep resistance compared to unreinforced Ti_3_SiC_2_ [17,18]. The unique properties of MAX/SiC composites predispose them to many applications, i.e., bearings, cylinders, gears, gas turbine seals, and rotational parts in engines [3,4].

This work aimed to investigate the influence of the addition of SiC particles to the Ti_3_SiC_2_ matrix on the mechanical properties of the obtained composites. According to our knowledge, this is the first work that comprehensively describes the entire process of producing composites based on Ti_3_SiC_2_ reinforced with SiC, starting from the synthesis of the MAX phases, the influence of the synthesis parameters, and the composition of the mixtures on the phase composition, to the consolidation of the composites.

## 2. Materials and Methods 

For the preparation of both the MAX-Ti_3_SiC_2_ phases and Ti_3_SiC_2_ matrix composites reinforced with SiC, commercial powders were used, the parameters of which are listed in Table 1.

Two carbon sources were used to synthesize the MAX-Ti_3_SiC_2_ phases: carbon black (CB) and synthetic graphite (SG). Both powders were mixed with titanium and silicon in two molar ratios: 3:1:2 and 3:1:3, respectively, for Ti:Si:C using a ball-type mill (Fritsch Pulverisette, Fritsch, Idar-Oberstein, Germany) in isopropyl alcohol suspension (no. 1759, Stanlab, Lublin, Poland). After drying, the powder mixture was sieved (# 300 µm) and synthesized using an SPS furnace (FCT Systeme GMBH, Effelder Rauenstein, Germany) in a graphite die. The synthesis parameters were as follows: temperature 1400 °C, heating rate: 250 °C/min, vacuum atmosphere (*p* = 5 × 10^−2^ mbar). Two schemes of the synthesis process were used. In the first, after reaching the maximum temperature, the sample was cooled, and in the second, the sample was annealed for two hours. After synthesis, the MAX phase was ground with an automatic mortar grinder (Retsch KM100, Retsch GmbH, Haan, Germany) below 45 µm (grinding bowl speed = 70 rpm, applied force 5–12.5 daN).

Ti_3_SiC_2_ powders selected as optimal at the first stage were used as a matrix to produce Ti_3_SiC_2_/xβ-SiC composites (where x = 2.5, 5, 7.5, 10 vol.%). Prepared powders mixtures were wet blended in a ball-type mill in Propan-2-ol with alumina grinding balls. After drying, the powders were sieved (# 300 µm). Composites were sintered with the SPS method with the following parameters: sintering temperature: 1300 and 1400 °C, heating/cooling rate: 250 °C/min, 4 min dwell time, 50 MPa, and vacuum. After the sintering process, cylindrical samples with a diameter of 20 mm and a thickness of 6 mm were obtained. Moreover, as a reference sample, pure Ti_3_SiC_2_ sinters were prepared. After sintering, specimens were mechanically polished down to a grit size of 0.2 μm and subjected to further investigations.

Both powders after the synthesis process and consolidated composites were subjected to phase analysis using an XRD diffractometer (Bruker D8 ADVANCE X-ray diractometer, Bruker Corporation, Billerica, MA, USA) with radiation Cu Kα (λ = 0.154056 nm). The parameters of this test were as follows: voltage: 45 kV, current: 40 mA, angular range: 10–155 deg with step 0.03. The composite microstructure, crack propagation, and powder morphology observations were performed on a scanning electron microscope (SEM Hitachi 5500, Hitachi, Tokyo, Japan). The observations were carried out at a 20 kV accelerating voltage. Moreover, the density and hardness of Ti_3_SiC_2_ matrix composites were examined using the Archimedes method (PN-EN 1389:2004 Standard) and Vickers Hardness Tester (FV-700e, Future-Tech, Kawasaki, Japan) using the indentation method under the load of 49.05 N, respectively. In addition, fracture toughness was measured with the indentation method using the Vickers Hardness Tester (FV-700e, Future-Tech, Kawasaki, Japan) under a load of 98.07 N.

## 3. Results

Figure 1, Figure 2, Figure 3, Figure 4 and Figure 5 show the MAX Ti_3_SiC_2_ phase synthesis process results. Two different carbon powders and different molar ratios of individual powders were used. In addition, in the last stage, the effect of annealing the powders after the synthesis process on the morphology and phase composition of the obtained MAX phases was checked.

Figure 1a–c show the results obtained for the powder synthesized with the molar fraction of 3Ti-1Si-2SG using synthetic graphite. The obtained powders are characterized by the presence of particles with equiaxed morphology and flake-shaped particles in which the layered structure typical for the MAX phases is visible (Figure 1a). Chemical analysis (Figure 1b) of individual particles showed the presence of Ti and Si (analysis 1) in equiaxed particles and Ti, Si and C (analysis 2) in flake particles, which indicates the presence of phases other than Ti_3_SiC_2_. This is confirmed by the phase analysis of the powders produced (Figure 1c). The presence of two dominant phases, Ti_3_SiC_2_ and TiC, and TiSi_2_, Ti_5_Si_3_, and residual amounts of unreacted graphite were detected. It follows that the equiaxed particles containing Ti and Si are TiSi_2_, and Ti_5_Si_3_, while the particles showing layered structure containing Ti, Si, and C are particles of the Ti_3_SiC_2_ phase.

Figure 2a–c show the test results of powders also synthesized with synthetic graphite, with its excess content (3Ti-1Si-3SG). The morphology of powders with an excess proportion of synthetic graphite differs significantly from the stoichiometric ratio. The powder particles show an irregular shape (Figure 2a). Additionally, many octahedral particles were observed that are typical of TiC [19]. Chemical analysis (Figure 2b) performed on these particles showed the presence of mainly Ti and C (analysis 1). In addition, the phase composition analysis confirms significant differences in relation to the equilibrium molar fraction of the powders. A much higher peak intensity was observed for TiC (Figure 2c). Ti_3_SiC_2_ is still present, but its peak intensity is much lower. In addition, high-intensity peaks from SiC and low-intensity peaks from TiSi_2_ were observed. These results confirm the presence of a large amount of octahedral TiC particles.

The research results of the powder synthesized with the 3Ti-1Si-2CB molar ratio and the use of carbon black as the C source are shown in Figure 3a–c.

The obtained research results are very similar to those achieved for powders synthesized with the same molar ratio and synthetic graphite used. The powder morphology also contains equiaxed particles and flake particles (Figure 3a). Moreover, the flake particles show a layered structure and consist of Ti, Si, and C as the main element (analysis 1, Figure 3b). In addition, the phase analysis showed the presence of the same phases in which Ti_3_SiC_2_ and TiC are dominant. In the case of powders in which CB was used as a carbon source and the 3Ti-1Si-3CB molar ratio, the morphology of powder particles is mostly similar to flake with also visible larger equiaxed particles (Figure 4a). However, the flake particles do not exhibit a layered structure. Chemical analysis of these flakes showed the presence of Ti, C, and Si (Analysis 1, Figure 4b). The phase composition of these powders is very similar to those obtained for SG, and the same molar ratio where the highest peak intensity was noticed for TiC. A much lower amount of octahedral TiC particles was observed for these powders.

For the next stage of Ti_3_SiC_2_ phase synthesis, powders with a stoichiometric molar ratio, and SB as a carbon source were used. The choice of the stoichiometric molar ratio was related to the fact that for these powders, the dominant content of the Ti_3_SiC_2_ phase was obtained. On the other hand, the choice of the carbon source was associated with the lack of significant differences in the phase composition of the synthesized powders with different forms of carbon and the lower price and greater availability of synthetic graphite. The selected powders after the synthesis were not cooled but were heated for 2 h at 1400 °C without applying pressure.

The research results of powder after the annealing process are shown in Figure 5a,b. The application of the annealing process radically changed the morphology of the powders. Powders with a layered structure typical of the MAX phases were obtained (Figure 5a). The phase analysis of the powders showed the presence of Ti_3_SiC_2_ as the dominant phase (Figure 5b). In addition, TiC, SiC, and TiSi_2_ have also been identified. These powders were used to produce composites based on them.

Two series of composites were produced, sintered at 1300 and 1400 °C using SPS, containing 2.5, 5, 7.5, and 10% SiC by volume. In addition, two reference sinters of pure Ti_3_SiC_2_ were prepared at the same temperatures. The results of apparent density measurements are shown in Figure 6. The relative density was not determined because the calculations could be burdened with a significant error resulting from the presence of several phases in the initial powder of the MAX phases and the possible decomposition of Ti_3_SiC_2_ during the consolidation process. In the case of reference samples, no effect of the sintering temperature on the density of the sinters was observed. Both sinters have almost identical densities above 4.5 g/cm^3^. Much more significant differences are visible in the case of composites. An increase in the sintering temperature significantly increases the density of the sinters. Composites sintered at 1400 °C show density practically at the same level as reference samples. In addition, no effect of the SiC content on the density of the obtained sinters was observed. This effect may be related to the possible decomposition of SiC particles during the sintering process. SiC particles can undergo surface degradation to TiC (which has a much higher density of 4.93 g/cm^3^) [18].

The results of composite density measurements confirm the observations of the microstructure. Figure 7a–d show examples of microstructures of composites reinforced with 7.5% SiC sintered at 1300 and 1400 °C. A similar microstructure (Figure 7a,c) containing elongated particles typical of the MAX phases was observed for both series of composites. Observations under higher magnification reveal much higher porosity in the case of composites sintered at 1300 °C (Figure 7b,d). In addition, in composites sintered at 1400 °C, octahedral particles’ presence (Figure 7d) was noticed, which in the case of powders were identified as TiC. The produced composites were also subjected to phase analysis using the XRD method. The test results of composites reinforced with 10% SiC sintered at both temperatures are shown in Figure 8a,b.

Analyzing the obtained diffraction patterns, significant differences between the samples sintered at 1300 and 1400 °C are visible. Composites sintered at 1300 °C show the same phase composition as the starting powder (Figure 8a). Only higher intensity of peaks from TiC can be observed. Increasing the sintering temperature by 100 °C changes the phase composition of the obtained composites. The appearance of another Ti_5_Si_3_ phase was observed (Figure 8b). Additionally, the peaks from TiC showed the highest intensity, which indicates the possible occurrence of the MAX phase decomposition during the sintering process.

Figure 9 presents the results of hardness tests of two series of composites. The results of hardness measurements show a similar trend as density measurements. Pure Ti_3_SiC_2_ shows similar hardness regardless of the sintering temperature. This is related to the density of the sinters. Regardless of the temperature, the sinters show almost identical density, which proves that the sintering temperature of 1300 °C is sufficient to obtain dense sinters. On the other hand, the temperature of 1400 °C is too low for significant changes in the phase composition. This translates directly into the obtained hardness measurement results. Much more significant differences are visible in the case of composites. Composites sintered at 1300 °C show a very low hardness of a maximum of 450HV for a composite containing 10% SiC, which confirms that they are poorly consolidated. Composites sintered at 1400 °C show a hardness higher than the reference samples, and the hardness slightly increases with the increasing reinforcing phase content.

Figure 10 presented the results of fracture toughness measurements as a function of the SiC volume fraction of composites sintered at 1400 °C. The measurements of composites sintered at 1300 °C were impossible due to the too-low level of sample consolidation. The presented results show that the increase in the content of the reinforcing phase causes an increase in fracture toughness. The lowest value was observed for the composite containing 2.5% SiC, almost identical to the reference samples. The highest K_IC_ value was observed for samples with the highest content of the reinforcing phase (7.5 and 10%) and was 6.3 MPa·m^0.5^.

Figure 11a–e presents images of crack propagation in Ti_3_SiC_2_/SiC composites. The cracks were made with a Vickers indenter. The produced composites are characterized by mixed crack propagation mechanisms typical of both granular materials and laminates. Thus, we can observe a zig-zag crack propagation that bypasses the particles and propagates through boundaries (typical for granular materials, Figure 11c) [20,21]. On the other hand, many more mechanisms typical for laminates can be observed, such as delamination (Figure 11a), layer bridging and layer cracking (Figure 11b,e), or layer debonding (Figure 11c). These mechanisms are related to the matrix material’s structure consisting of alternating layers with strong ionic bonds, acting as fibers, separated by layers with weak metallic bonds [22,23]. Figure 11f shows the Vickers indenter imprint. This is a typical imprint for a ceramic material with cracks propagating from the corners.

## 4. Discussion

Phase analysis and morphology observations of the obtained powders confirm the possibility of synthesizing relatively pure Ti_3_SiC_2_ powders by the SPS method (Figure 1, Figure 2, Figure 3 and Figure 4). The conducted tests showed no influence of the form of carbon on the phase composition of the obtained MAX phases. The use of CB due to the smaller particle size and, thus, higher reactivity should translate into the possibility of obtaining a different phase composition than for SG. Similar studies were conducted by Tabares et al. [24]. However, they showed that phase purer Ti_3_SiC_2_ powders could be achieved using carbon with a larger particle size. They explained these results by the fact that in the case of syntheses with Ti, SiC, and C, carbon reacts mainly with titanium to form TiC, and its reactivity is not as important as in the case of the Ti, Si, C system. However, despite using pure elements, no effect on particle size was observed in our study. This may be due to the additional effect of particle agglomeration during powder mixing. The smaller the particle size, the greater the tendency to agglomeration and the more difficult it is to obtain a homogeneous distribution. Therefore, as a consequence, taking into account the presence of agglomerates, the effect of finer particles can be negligible. Significantly more significant differences in the phase composition of the synthesized powders can be observed for different molar ratios. In the case of 3Ti-1Si-2C, all the titanium reacted to form Ti_3_SiC_2_, TiC, TiSi_2_, and Ti_5_Si_3_. The Ti_5_Si_3_ and TiC phases are the main intermediates in forming the MAX phases [24]. However, TiC can also be the result of the decomposition of the MAX phases. In addition, there is also unreacted graphite, which may indicate its excessive content in the initial mixture of powders. Such a phase composition may suggest the need to anneal the powder after the synthesis process, during which TiC and Ti_5_Si_3_ could react with the other components to form the MAX phase. This is confirmed by the results presented in Figure 5. The annealing process caused the graphite and Ti_5_Si_3_ to disappear and significantly increased the intensity of the peaks from the MAX phases (Figure 5). Consequently, the obtained phase composition is typical of that obtained by other authors using various methods [25,26,27,28,29,30]. Using an excessive amount of carbon (3Ti-1Si-3C) changes the phase composition of the produced powders. Ti_5_Si_3_ and unreacted carbon are no longer observed. However, SiC is formed, and the intensity of the TiC peaks is much higher than in the case of powders with an equilibrium amount of carbon. The use of annealing should theoretically ensure the formation of MAX according to Reaction (1):(1)$$TiC+2TiSi_{2}+SiC\;\to \;Ti_{3}SiC_{2}+4Si$$
however, due to the high Gibbs free energy, these reactions are not favorable at 1400 °C [24]. Changes in the phase composition of the obtained powders are also visible based on the morphology of the powders, in the case of powders containing the most Ti_3_SiC_2_, the content of layered particles typical of the MAX phases increases.

As a result of the sintering of the mixtures of Ti_3_SiC_2_/SiC powders, consolidated composites were obtained. The results show that the sintering temperature of 1300 °C is too low for composites reinforced with SiC (Figure 6). It is also confirmed by the microstructure observations where numerous voids at the boundaries of individual particles were noticed (Figure 7). A much higher density, close to the density of reference samples, was observed for composites sintered at 1400 °C. However, when comparing the phase analysis results, it can be seen that the higher sintering temperature causes changes in the phase composition of the composites (Figure 8). For sinters at 1300 °C, the phase composition is almost identical to that of the initial powder, which proves the lack of degradation of the MAX phases at this temperature. Increasing the temperature by 100 °C resulted in an additional Ti_5_Si_3_ phase and an increase in the intensity of the TiC peaks. This proves the ongoing degradation processes of the MAX phases at this temperature. On the other hand, the increase in temperature and the degradation processes resulted in a higher degree of composite consolidation, which is shown by the observations of the microstructure and the measurements of the density of sinters. The results are different from those in the literature, where the appearance of Ti_5_Si_3_ was observed at temperatures as low as 1200 °C [13]. This may be due to the presence of SiC or the use of a fast SPS sintering process. The low consolidation level of composites sintered at 1300 °C influenced very low results of composite hardness measurements and the inability to perform K_IC_ measurements using the indentation method (Figure 9 and Figure 10). In the case of composites sintered at 1400 °C, a significant strengthening effect associated with the addition of SiC is visible. Both hardness and K_IC_ increase with increasing SiC content. The obtained hardness is even higher than that which can be found in the literature for composites reinforced with 20 vol. % of SiC [13].

The observations of crack propagation in the obtained composites are extremely interesting (Figure 11). The occurrence of crack propagation mechanisms typical for granular materials and laminates is observed. As mentioned, such cracking mechanisms as zig-zag crack propagation, delamination, layer bridging, layer cracking, or layer debonding can be observed. The presence of mechanisms typical for laminates is extremely beneficial for these materials because these mechanisms absorb large amounts of fracture energy. This is related to the significant lengthening of the cracking path or blocking its propagation by bridging, which limits the possibility of cracking propagation into the material.

## 5. Conclusions

The paper presents the research results on the development of MAX Ti_3_SiC_2_ powders. The test results show that the highest purity of the powders was obtained for the molar fraction of 3Ti-1Si-2C using the two-hour heating process at 1400 °C. In addition, no effect of the carbon form on the phase composition of the produced MAX phases was observed. Moreover, composites reinforced with different content of SiC particles were sintered with the produced MAX phases. The obtained test results showed that it is possible to consolidate composites characterized by high density and mechanical properties. In addition, the occurrence of crack propagation mechanisms typical of both granular materials and laminates was observed.

## Figures and Tables

**Figure 1 materials-16-00889-f001:**
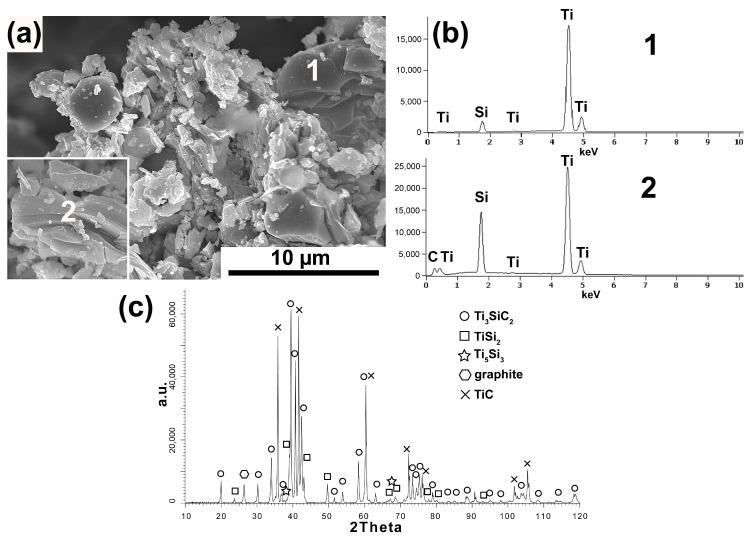
Morphology (**a**), chemical analysis (**b**), and phase analysis (**c**) of Ti_3_SiC_2_ powder synthesized with molar ratio 3Ti-1Si-2SG.

**Figure 2 materials-16-00889-f002:**
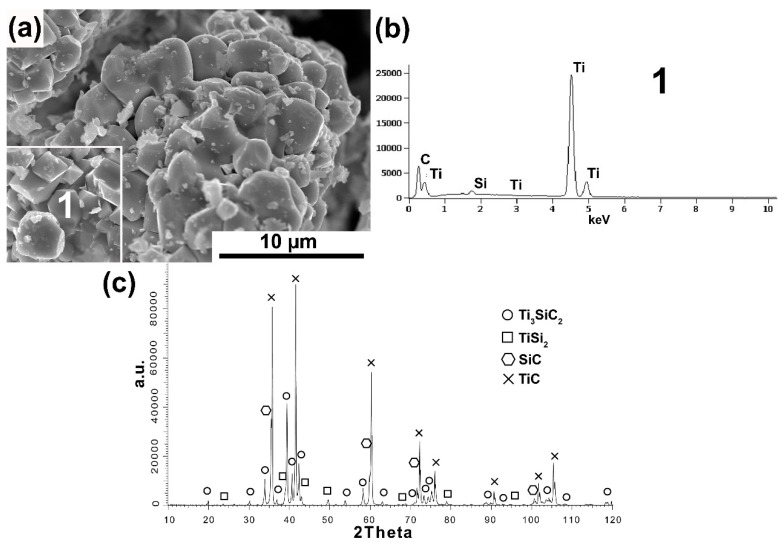
Morphology (**a**), chemical analysis (**b**), and phase analysis (**c**) of Ti_3_SiC_2_ powder synthesized with molar ratio 3Ti-1Si-3SG.

**Figure 3 materials-16-00889-f003:**
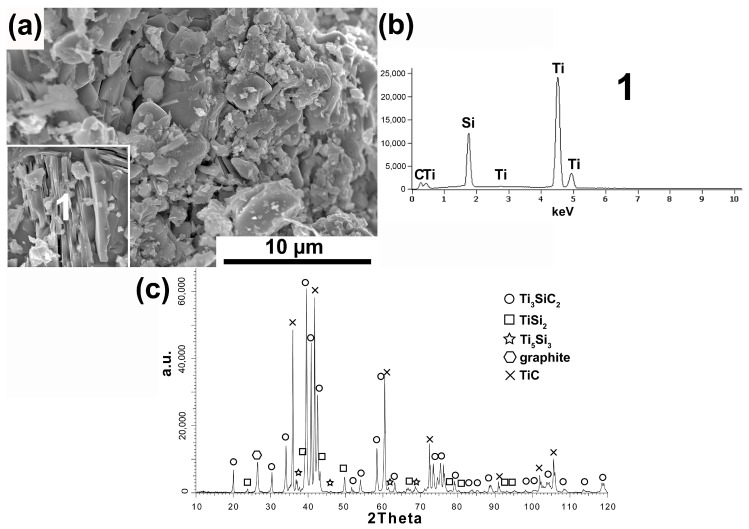
Morphology (**a**), chemical analysis (**b**), and phase analysis (**c**) of Ti_3_SiC_2_ powder synthesized with molar ratio 3Ti-1Si-2CB.

**Figure 4 materials-16-00889-f004:**
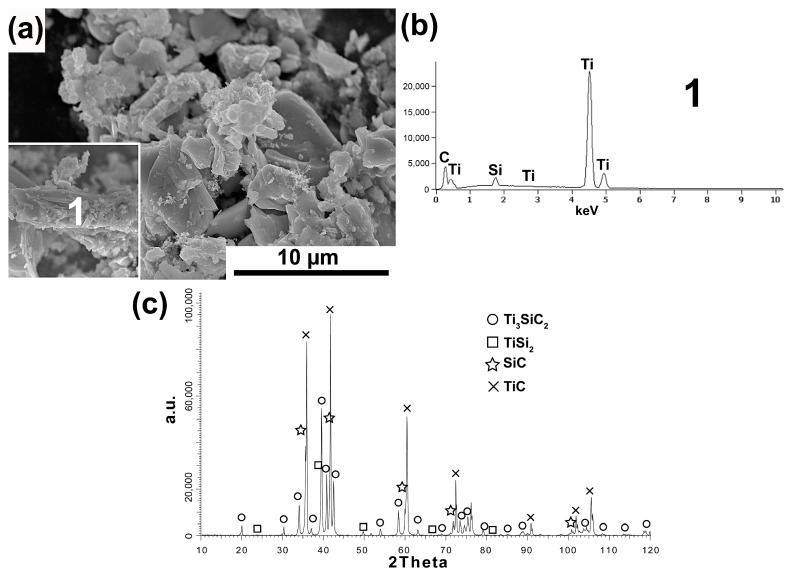
Morphology (**a**), chemical analysis (**b**), and phase analysis (**c**) of Ti_3_SiC_2_ powder synthesized with molar ratio 3Ti-1Si-3CB.

**Figure 5 materials-16-00889-f005:**
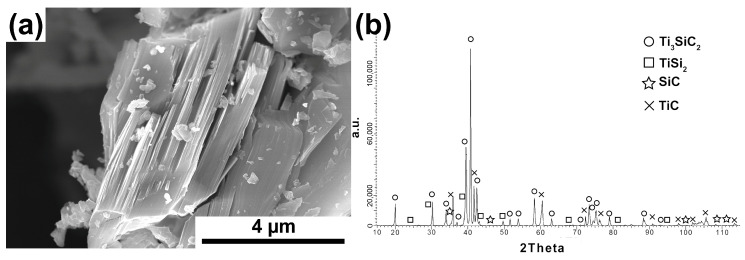
Morphology (**a**), and phase analysis (**b**) of Ti_3_SiC_2_ powder synthesized with molar ratio 3Ti-1Si-2SG annealed for 2 h.

**Figure 6 materials-16-00889-f006:**
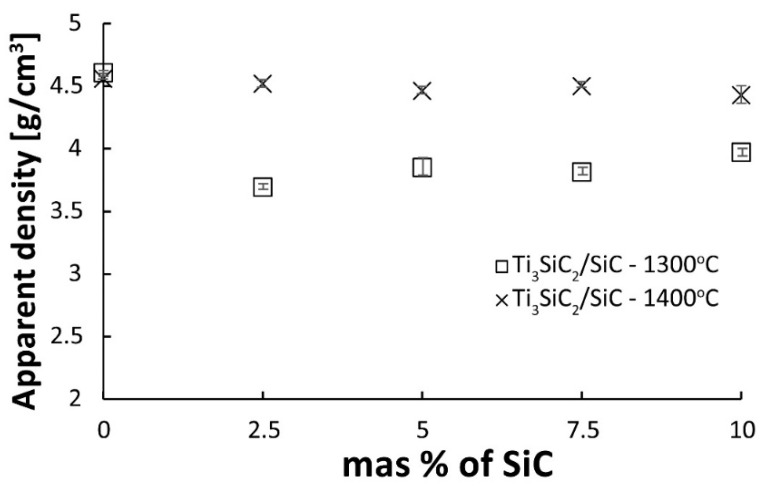
The apparent density of Ti_3_SiC_2_/SiC composites sintered at 1300 and 1400 °C.

**Figure 7 materials-16-00889-f007:**
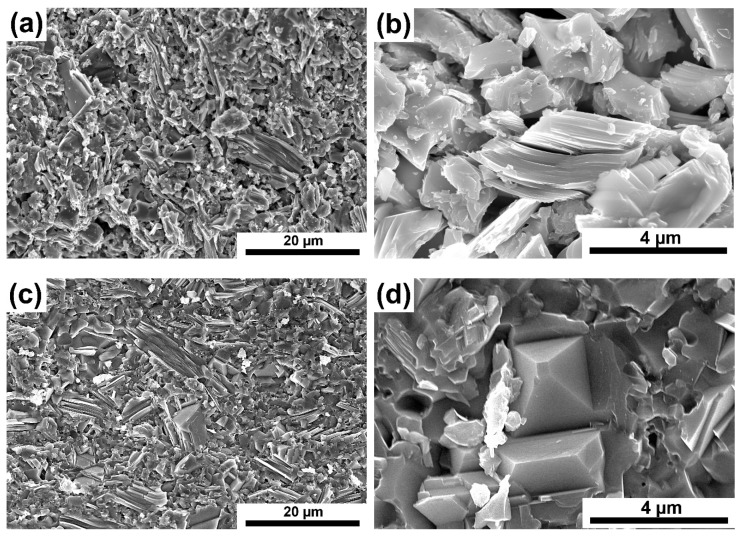
Microstructure of the Ti_3_SiC_2_ matrix composites reinforced with 7.5 vol. % SiC sintered at 1300 °C (**a**,**b**), and sintered at 1400 °C (**c**,**d**).

**Figure 8 materials-16-00889-f008:**
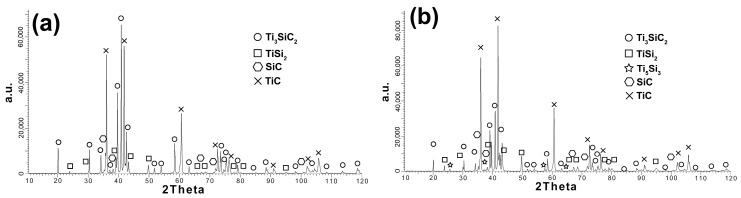
Phase analysis of the Ti_3_SiC_2_ matrix composites reinforced with 10 vol. % SiC sintered at 1300 °C (**a**), and sintered at 1400 °C (**b**).

**Figure 9 materials-16-00889-f009:**
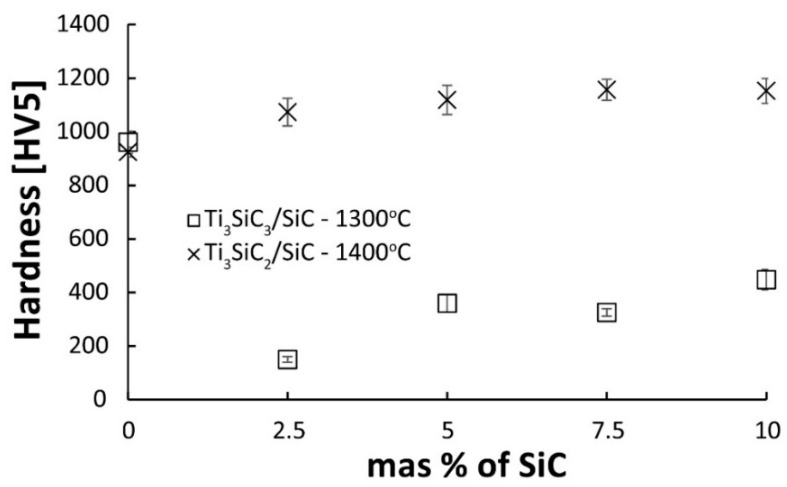
The Vickers hardness of Ti_3_SiC_2_/SiC composites sintered at 1300 and 1400 °C.

**Figure 10 materials-16-00889-f010:**
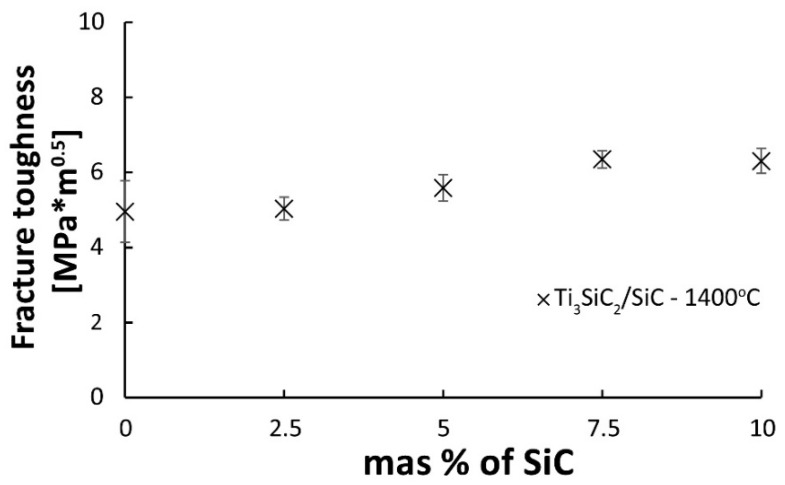
The fracture toughness of Ti_3_SiC_2_/SiC composites sintered at 1400 °C.

**Figure 11 materials-16-00889-f011:**
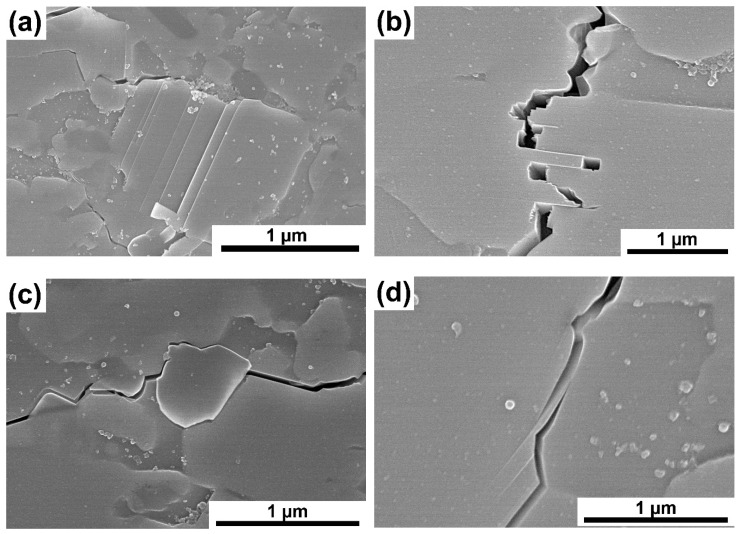
Fracture propagation of the Ti_3_SiC_2_—7.5 vol. % SiC sintered at 1400 °C (**a**–**d**), and (**e**,**f**) Vickers indenter imprint.

**Table 1 materials-16-00889-t001:** Parameters of the powder substrates used.

Powder	Purity	APS *	Manufacturer
Titanium	99.6%	<20 µm	GoodFellow, Cambridge, UK
Silicon	99%	<5 µm	Atlantic Equipment Engineers, Upper Saddle River, NJ, USA
Synthetic graphite	99.9%	<20 µm	Sigma-Aldrich, St. Louis, MO, USA
Carbon black	99%	<100 nm	Sigma-Aldrich, St. Louis, MO, USA
Silicon carbide	99.8%	0.42 µm	Alfa Aesar, Ward Hill, MA, USA

* Average particle size.

## Data Availability

The data presented in this study are available on request from the corresponding author.

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
