# Peer review of "Synthesis of Ti_3_SiC_2_ Phases and Consolidation of MAX/SiC Composites—Microstructure and Mechanical Properties"

_materials, 2023, doi:10.3390/ma16030889_

Round 1

Reviewer 1 Report

The following items should be considered by the authors.

1-      In the abstract, it is necessary to state the most important results quantitatively.

2-      The introduction is well written. However, the innovation as well as the difference between the present manuscript and available studies should be clearly explained at the end of the introduction.

3-      The dimensions of the prepared composite samples should be determined.  Cylindrical samples? What dimensions?

4-      Line 154: “Ti2SiC2” or “Ti3SiC2”?!!

5-      Line 195, 196: “In addition, no effect of the SiC content on the density of the obtained sinters was observed”. Since the density of silicon carbide (3.21 g/cm3) is less compared to the matrix, it is expected that the density will decrease with increasing the amount of SiC. Why is this issue not visible in the results of Figure 6?

6-      Line 190-192: “In the case of reference samples, no effect of the sintering temperature on the density of the sinters was observed. Both sinters have almost identical densities above 4.5 g/cm3”.  Also, line 225-226: “Pure Ti3SiC2 shows similar hardness regardless of the sintering temperature”. Why? Reason? Add an explanation.

7-      The magnification of figures 7(b) and 7(d) is not the same. For this reason, it becomes difficult to compare and check the presence of porosity.

Author Response

The answer is attached in the file.

Reviewer 2 Report

In this paper, the work done to synthesis Ti3SiC2 powder by spark plasma sintering process. The investigation was carried out to study the influence of the addition of SiC particles to the Ti3SiC2 matrix on the mechanical properties.

The steps taken for the preparation method were well defined. The results, analysis, and discussion were explained well, proving the worthiness of the manuscript. The cited references were mixed from old and recent works.

English usage in the manuscript requires some revision. A glaring grammatical, and structural errors can be seen throughout the draft. And for temperature values, there should be a space between the value and the unit for oC.  

Suggest revising Abstract section as the present form is quite general and not summarize well the details content of the manuscript, especially the most significant parameter values from the results needed to be highlighted.

Author Response

The answer is attached in the file.

Reviewer 3 Report

The submitted manuscript addresses the synthesis of Ti3SiC2 powder and its application for the manufacturing of MAX/SiC composites. Worthy of particular mention are the microstructural analyses accompanying all steps  of the MAX phase synthesis and manufacture of the MAX/SiC composites.

There are some terminological discrepancies within the submitted manuscript such as:

- What does the expression "two different forms of carbon" mean? Different sources, different morphologies, different phase composition or different particle sizes etc? 

- The term "Optimization" indicates rather an industrious work than a methodological development work. 

- The European standard EN 1094-4 is intended for insulating refractory products, i.e. materials with a high porosity. 

- The term "apparent density" (Figure 6) implies the absence of open porosity.  Please verify if bulk or apparent density was measured using Archimedes principles. Did the authors determine the true density of the synthesized MAX phases by means of gas (helium) pycnometry?

- Some sentences such as "The results of hardness measurements are identical to the measurement of the density" (page 9, line 224-225) must be rewritten. Please check the whole manuscript for imprecise phrasing and inconsistencies.

Furthermore, following issues are missing:

- scope of application area of the MAX/SiC composite

- specification of the composite sample´s dimensions

- information on the preparation of the composite sample´s surface prior the hardness measurements

- SEM micrographs of the composite surface after the Vickers hardness identation in order to take a closer look at the whole indent area left by the Vickers pyramid

Author Response

The answer is attached in the file.
